# Effects of an Exercise Program and Cold-Water Immersion Recovery in Patients with Rheumatoid Arthritis (RA): Feasibility Study

**DOI:** 10.3390/ijerph20126128

**Published:** 2023-06-15

**Authors:** Daniele Peres, Clément Prati, Laurent Mourot, Amanda Magalhães Demartino, Yoshimasa Sagawa, Nicolas Tordi

**Affiliations:** 1PEPITE EA4267, Platform Exercise Performance Health Innovation (EPHI), Franche-Comté University, F-25000 Besançon, France; cprati@chu-besancon.fr (C.P.); nicolas.tordi@univ-fcomte.fr (N.T.); 2Rheumatology Department, CHRU Besançon, F-25000 Besançon, France; 3EA3920 Prognostic and Regulatory Factors of Cardiac and Vascular Diseases, Exercise Performance Health Innovation (EPHI), Franche-Comté University, F-25000 Besançon, France; laurent.mourot@univ-fcomte.fr (L.M.); amanda.magalhaes_demartino@univ-fcomte.fr (A.M.D.); 4Integrative and Clinical Neurosciences EA481, Inserm 1431, Franche-Comté University, F-25000 Besançon, France; ysagawajunior@chu-besancon.fr

**Keywords:** aerobic exercise, rheumatoid arthritis, cardiovascular disease, cryotherapy

## Abstract

Rheumatoid Arthritis (RA) patients present is an increased cardiovascular risk (CVR) linked to systemic inflammatory manifestations. A physical activity program with known positive effects on CVR, followed by cryotherapy because of its analgesic and anti-inflammatory effects, may be interesting. However, there are no reports in the literature of such a program. This study aimed to determine the feasibility (acceptability, safety, and effectiveness) of an individualized Intermittent Exercise Program followed by cold-water immersion as a recovery for RA patients. The program was conducted three times per week by eighteen RA patients (one man) with means of age and BMI of 55 (11.9) years and 25.5 (4.7) kg·m^−2^. Outcomes were assessed before and after nine and seventeen sessions and included evaluation of acceptability by perceived exertion (Borg) and water temperature (VAS) measures at each session; safety by a number of painful and swollen joints (echography); physical function (health assessment questionnaire); general health status (Short Form-36) measures; and effectiveness by arterial stiffness (pulse wave velocity, or PWV) measures. The results showed good acceptability of the program; no patient dropped out of the protocol or even presented difficulties or perceived pain. The HR and PWV values decreased significantly (70.2 ± 8.4 to 66 ± 5.5; *p* < 0.05 and 8.9 ± 1.2 to 7.0 ± 0.8; *p* < 0.001) after nine exercise sessions. No aggravation of symptoms has been noted. This program is acceptable, safe, and effective; consider tailoring it for supervised home-based use.

## 1. Introduction

Rheumatoid arthritis (RA) is an autoimmune disease presenting, besides the joint pain, stiffness, and swelling, a risk for cardiovascular disease twofold higher than the general population and nearly threefold higher in patients with 10 years or more of disease [1,2,3]. This risk is independent of traditional risk factors (e.g., diabetes, hypercholesterolemia) and assigned to immune dysregulation, sustained inflammation, and recurrent use of certain pharmacological agents (e.g., corticosteroids) [4]. These factors facilitate hypertension, arterial stiffness [5], and atherosclerosis early and critically [4], and are all considered cardiovascular risk (CVR) factors [6]. In addition, the patients still present a sedentary lifestyle, a known risk factor for cardiovascular disease, which is therefore exacerbated when added to disease-related risk factors [7,8]. In such a context, physical activity is a positive element in combating the factors of CVR [9].

However, RA patients are often unfit and less active than the general population. This is precisely the obstacle to be overcome in the rehabilitation of patients with RA. A recent study showed that the RA population presents a fear of movement exacerbating disease activity [10], which partly explains the physical inactivity of this population and the difficulty of managing exercise at an appropriate intensity to minimize CVR without increasing disease symptoms [7,8].

Information about the benefits and potential risks of physical activity, as well as an initial follow-up with a professional, can reassure the patient about this practice. In addition, proposing physical activity with well-controlled intensity, without joint impact, and followed by adjuvant therapy, such as cryotherapy, may be an interesting strategy to motivate the patient. The analgesic and anti-inflammatory effects of cryotherapy have been described in rheumatic patients [11,12,13]. However, there is no protocol associating a physical activity program with the application of cryotherapy in patients with RA described in the literature [14]. Cold water immersion seems to be the most favorable cryotherapy method to be used because it is low-cost, simple, and feasible for the patient himself at home.

In this context, the main objective of the present study was to investigate the feasibility, acceptability, and safety of an individualized interval exercise program followed by cold-water immersion as a recovery for a single group of RA patients. Feasibility and acceptability were assessed in terms of adherence and tolerance to exertion and water temperature. The safety of the intervention was tested in terms of no adverse events, no increase in the number of painful and swollen joints, and well-being in terms of activity limitations and quality of life level. The effectiveness of the program to impact arterial stiffness was evaluated by pulse wave velocity measurement prior to and after nine and seventeen exercise sessions, respectively.

So, it was hypothesized that (a) administering an exercise intervention followed cold water immersion would be safe and not associated with any adverse events; (b) exercise would have an uptake and adherence rate of greater than 85%; (c) a trend toward improvements in the cardiovascular system would be found in the patients; and (d) a trend toward improvements in the activity disease (inflammation, pain) by those who went on to phase II.

## 2. Materials and Methods

### 2.1. Participants 

A convenient sampling of RA patients (male and female) was recruited in service at the University Hospital (CHRU) in Besançon. They had a thorough diagnostic assessment instituted prior to recruitment and were identified as eligible if they were between the ages of 20 and 70 years and were diagnosed with rheumatoid arthritis with a DAS-28 score between 2.6 and 6 points. Participants were excluded if they had: unstable use of corticosteroids and/or >10 mg prednisone/day and/or hypertension (>140 systolic blood pressure); pregnant women; alteration of higher functions making comprehension and adherence to a conditioning program impossible; contraindication to immersion in cold water (a dermatological factor and/or vascular or respiratory cardiac dysfunction and/or cold intolerance syndrome); inability to perform exercise, regardless of its origin (neurological, central or peripheral and/or legal incapacity or limitation). There were no restrictions regarding BMI or fitness level (VO_2_, energy expenditure). The participants neither changed nor stopped their pharmacological treatment during the protocol. Ethics approval for the study was obtained from the Ile-de-France VII Research Ethics Committee (2017-AO1422-52), and all participants provided informed written consent to participate in the study.

### 2.2. Intervention

This study proposes an interval exercise program followed by cold water immersion recovery. The program was divided into two phases, each lasting 3 weeks (see Figure 1). The sessions were separated by at least 48 h (3 sessions/week). The total duration of the session was approximately 45 min, with an addition of 30 min on evaluation days (S1, S9, and S17, for a total duration of 75 min) (see Figure 1). All sessions were individual and supervised by a physical activity professional, and the patients were informed about the program and the effects of physical activity. Each participant, according to his availability, selected the same period of the day to carry out all the sessions. They were also encouraged in each session. All sessions were performed on the EPHI platform in Besançon, France.

#### 2.2.1. Interval Exercise Program 

Based on the international guidelines and evidence for RA [15,16], the participants realized a pedaling exercise for 6 consecutive periods of 5 min, including a 4-min “base” work followed by a 1-min “peak” [17] (total of 30 min). The base period was performed at a moderate intensity set at around 60% of the Heart Rate Reserve (HRR). During the peak period, the participant must reach 80% of HRR. The average total heart rate was 72% of HRR. The workloads of each period were increased by 10 watts each time the heart rate targets were not reached, starting with the base period followed by the peak period.

#### 2.2.2. Water Cold Immersion

The cryotherapy for recuperation was applied immediately after pedaling exercise for 15 min with both legs immersed in cold water up to waist-deep inside an inflatable pool (Cryo Control Team, Toulouse, France). The water temperature was maintained at 15 ± 0.3 °C.

The echography and demographic data (body mass index, tobacco consumption, professional status, drug treatment, and PA level assessed by the SQUASH questionnaire [18]) were collected at baseline, after S9 and S17 in the hospital. The adherence, tolerance, safety, and effectiveness were collected in our laboratory, where the sessions take place (see Figure 1). An observation notebook was developed for the manual recording of all measurements, even those that were recorded on the computer or directly by the measuring instrument.

### 2.3. Measures 

#### 2.3.1. Adherence by Assiduity

At every session, the patients were asked about the pain during day-out sessions and to keep a diary of their exacerbations. The treatment alterations were registered. In the case of an exacerbated disease perceived by patients, they must stop the program, and if they do not recover enough to start participation within 1 week, they are classified as dropouts.

#### 2.3.2. Tolerance

Perceived exertion after exercise was assessed by a Borg scale between 6 (no exertion) and 20 (maximal exertion) [19]. A visual analog scale (VAS-water scale) [20] was used for the perception of water temperature tolerance on a Likert scale between 1 (totally tolerable) and 5 (totally intolerable) points. It was assessed after the first, fifth, tenth, and fifteenth minutes of immersion in water.

#### 2.3.3. Safety

Echography was performed to monitor or identify the number of inflammatory joints using an Esaote MyLab five aircraft on 32 joints of the hands, wrists, and feet using a B-score and Doppler synoptic mode, each from 0 to 32, for a total of 0 to 64 joints. The measurement was performed by one experienced doctor in clinical routine who was not informed of the results of the other measures. Additionally, the number of painful and swollen joints was counted from the participants’ reports. The VAS was a line where the ends were marked with no pain (0 points) or the most severe pain (100 points). The participants marked a point that matched the intensity of the pain they experienced. Questionnaires were applied with respect to the mode of employment described by the authors to verify the functional ability level through the Health Assessment Questionnaire (HAQ) [21] and the quality of life through the physical and mental composite of the 36-Item Short Form Survey (SF-36) [22].

#### 2.3.4. Effectiveness

The effectiveness of the program was tested by measuring Pulse Wave Velocity (PWV, m·s^−1^) in the central segment, blood pressure (systolic SBP and diastolic DBP, mmHg), and heart rate (HR, bpm). All effectiveness measures were taken in a supine position.

A validated and non-invasive measure of the PWV was realized by the Complior SP device (Artech Medical, France). The procedures were conducted according to the previous study [23]. The measurement was performed by an experienced evaluator who had no knowledge of the previous results of the study. At least 3 measurements were taken, and the median was used. Measurements with a signal quality lower than 85% were not accepted, as per instrument guidelines.

SBP, DBP, and HR were obtained using an automatic armband blood pressure monitor (Omron Healthcare Company, Kyoto, Japan) [24].

### 2.4. Sample Size Calculation

This is a non-comparative proof-of-concept test with an experimental arm. The measure of program effectiveness was used to calculate the sample size: the proportion of patients with RA for whom there is a decrease between the starting PWV value (at baseline) and the PWV measured at the end of 3 weeks of the program of 1.5 m·s^−1^ according to Fleming design [25]. This type of design (Fleming one step) involving a limited number of patients allows an exploratory way to stop early the test of the program if the results observed are below a minimum efficiency (stop for futility) or above a ‘good’ efficiency (stop for efficiency with a continuation of the program). To achieve these goals, it is necessary to include at least 18 patients [26]. The null hypothesis was rejected if the lower limit of the one-sided 95% confidence interval of the efficiency was greater than 0.33.

The primary objective of phase II clinical trials is generally to assess the “therapeutic effectiveness” of a specific treatment. Thus, it is interesting to conduct a smaller-scale study that can be employed in these trials to decide whether or not the therapeutic efficacy merits further investigation. For ethical reasons, once the results are inadequate, this should allow for the early termination of the study.

### 2.5. Statistical Analysis 

All analyses were performed using SigmaStat software (SPSS Inc., Chicago, IL, USA). Descriptive statistics were used to profile the study participants’ safety, adherence, tolerance, and effectiveness outcomes. The results are presented as mean ± SD. After verifying the normality using the Kolmogorov–Smirnov test, a paired *t*-test was used to compare the measures in phase I of the program between values obtained at baseline and S9. A one-factor ANOVA for repeated measures was performed to examine the effects of time among the values obtained at baseline, at the end of phase I (S9), and at the end of phase II of the program (S17). In cases of statistical significance, Tukey’s post hoc test was performed. A *p*-value of <0.05 was considered statistically significant.

## 3. Results

Eighteen participants completed phase I of the protocol; their characteristics are described in Table 1. All participants presented 100% assiduity in the nine sessions corresponding to phase I of the program and reported that the sessions were enjoyable, beneficial to their health, and that they would recommend them to others. The participants reported no adverse effects or exacerbations of the disease during the program.

The results of perceived exertion measured by the Borg scale showed an average that remained around 11–12 points, representing an exertion classified as between ‘light’ and ‘somewhat hard’ activity. The average water temperature tolerance evaluated by the VAS-water scale was in the mid-range and improved with immersion time (see Table 1).

All participants showed no significant decrease in the number of inflammatory joints, according to echography. According to the participants’ reports, 44.5% (eight participants) showed no significant decrease in the number of painful joints, 44.5% did not change, and 11% (two participants) showed no significant increase after session nine. In relation to the swollen joints, only four participants reported some swollen joints at baseline, and all of them showed a decrease after session nine. All participants showed no significant decrease in pain after S9 (see Table 1). The scores of the HAQ and the SF-36 questionnaires showed no significant change.

The measure of program effectiveness evaluated by the pulse wave velocity showed a significant decrease of 1.9 m/s from the baseline values. SBP, DBP, and HR showed no significant changes (see Table 1).

Among the eighteen participants, eleven showed a reduction in pulse wave velocity of at least 1.5 m/s after phase I of the program and therefore proceeded to phase II. All participants presented 100% effort in the eight additional sessions of phase II. The tolerance results did not differ from phase I. In relation to the safety and efficiency measures, there was no significant difference after phase II (S17) (see Table 2).

## 4. Discussion

The literature presents consolidated information about increased cardiovascular risk in the RA population compared to the general population [6]. Aerobic exercises are recommended for this population in order to diminish this risk [27]. However, the RA population is less active than the general population, and one possible explanation may be the fear of movement known as kinesiophobia [10]. An inappropriate program can have adverse effects on those patients most susceptible to inflammation and strengthen their belief that physical activity can be harmful. We designed a tailored interval training program, including cryotherapy by cold water immersion for its well-known analgesic and anti-inflammatory effects, to give the RA population an opportunity to safely and efficiently practice physical activity. Consequently, the present study evaluates the safety, tolerability, and efficacy of this exercise program dedicated to RA patients in order to fight sedentarism and cardiovascular risks.

Our results proved the feasibility and acceptability of the program since all participants followed the full number of sessions per week. Mainly, two reasons may explain this result. First, as revealed by the SQUASH responses, most of our participants were physically inactive [18], therefore the supervision of each session and encouragement given to the patient have been effective. Indeed, a recent review of the literature showed that encouragement from health professionals is particularly a facilitator for physical activity and exercise [28] when the patients are less active. Second, the intensity of each exercise as well as the increase in workload intensity were adapted and individualized. Interval aerobic exercise is admittedly an efficient tool to fight against cardiovascular risks or diseases [29]. Many different interval aerobic exercise programs are proposed for patients to fight the consequences of their diseases and sedentarism. Such programs, named PEP’C [30] or SWEET [17], characterized by short bouts of high intensity followed by moderate intensity, are widely used in protocols with different patients [31,32,33]. It is more interesting for rehabilitation precisely because it allows accumulating more vigorous exercise time in an interval manner than could be achieved during a single continuous exercise [33].

Because of their characteristics, the exercise sessions proposed herein have been well tolerated; the rate of perceived exertion corresponds to moderate exercise. Moreover, despite the discrepancy in age, BMI, tobacco consumption, and chronicity of the disease in our population, the standard deviation for perceived exertion stays low throughout the program duration.

The same occurred with the cold-water immersion; the temperature was well tolerated. The water temperature was intentionally chosen to not be extremely cold, comfortable, and easy to achieve in the patient’s home but still show evidence of benefits in post-exercise recovery. Yeung et al. (2016) reported less muscle soreness in participants who performed cold water immersion (12 °C to 15 °C) after exercise when compared to the control group [34].

Regarding safety measures, there was no exacerbation of inflammation when considering the results of the echography, the number of painful swollen joints, and the pain. There was no change in the pharmacological treatment of the patients that could explain the positive results in the levels of inflammation measured by ultrasound and the painful and swollen joints of these patients. Likewise, the results of the functionality and quality of life questionnaires showed no change. These questionnaires were applied to control the eventual negative impact that the protocol could have on physical function and physical and mental well-being.

Concerning the PWV values used as an effective measure of the program, first, the PWV values of our patients measured before the program (8.9 m·s^−1^) are higher than the normal reference PWV (5.8 m·s^−1^) values for the general population [35] and comparable to the PWV (9.1 m·s^−1^) previously published with an RA population of the same age [36].

Second, the type of exercise used in this study predisposes to greater cardiovascular alterations (acute improvement in PWV values ~1 m·s^−1^) according to previous studies in healthy subjects [37,38]. The oscillations in hemodynamic conditions during interval training cause variations in peripheral vascular changes that may trigger the release of endothelial vasoactive or metabolic factors and consequently have a greater effect on arterial stiffness [17,37].

Third, considering the great heterogeneity of the duration of the protocol found in the literature [39], this protocol was composed of two phases to verify its effectiveness in the short and medium term. Among the eighteen patients who participated in the protocol, eleven showed a decrease of at least 1.5 m·s^−1^ in the PWV value and proceeded to phase II. It can be seen that the effectiveness of the protocol is linked to the patient’s profile. The participants who went to phase II presented a higher average age, body mass index, chronicity of disease, echography, and PWV compared to the participants who did not proceed to phase II. So, this program was more effective for patients with more severe clinical conditions that tend to obtain better results in the short term.

Finally, although the protocol showed a reduction in PWV for all patients, it was more effective for a specific profile of patients. Patients who did not proceed to phase II showed an average improvement of 1.1 m·s^−1^ compared to baseline PWV values. Analyzing the characteristics of these patients, it was possible to identify a lower BMI and diagnostic time with a lower number of painful and swollen joints compared to the patients who went to phase II. Thus, it is conceivable to select the profile of the patients to ensure the effectiveness of the protocol. Because these patients are more resistant to the practice of physical activity, an initial follow-up with a health professional for proper guidance and information is a priority.

Regarding the dose–response, the highest percentage of improvement was obtained after nine sessions of the program. The results between the beginning and the end of phase II did not show a significant decrease in PWV values. Considering the age of our population, we can consider the values to be within the normal range. Thus, the continuation of the program would be recommended for the maintenance of the results.

This study is about the feasibility of a protocol and therefore presents some limits. We established our sample calculation based on effectiveness criteria, causing patients who did not progress to drop out of the study. However, these patients that we considered to have a less severe clinical status and to be less responsive in relation to the program could make progress if they had more phase II sessions. Moreover, our sample was homogeneous in relation to the RA patients, who present with a heterogeneous clinical condition with different age ranges and levels of inflammation, pain, treatments, etc. According to the literature, RA patients usually present with associated pathologies, such as hypertension. The majority of patients in our study present with normal blood pressure despite a high PWV. Considering the characteristics of our population, it is difficult to generalize our results to the RA population, which is much more heterogeneous. Even if this study demonstrated the feasibility and safety of our protocol, a larger population with various blood pressure statuses, durations of the disease, and levels of inflammation is necessary to better define the recommendations of this protocol.

## 5. Conclusions

In summary, this study demonstrated a feasible, acceptable, and safe form of a physical activity program followed cold-water immersion for people with RA. No patient dropped out of the protocol, even if they presented difficulties or perceived pain. No aggravation of symptoms, therefore a positive element to fight against kinesiophobia and the risks linked to sedentary life, is the highest PWV. These findings should provide for future projects applying for this program at home under the supervision of physiotherapists specialized in physically adapted activity.

## Figures and Tables

**Figure 1 ijerph-20-06128-f001:**
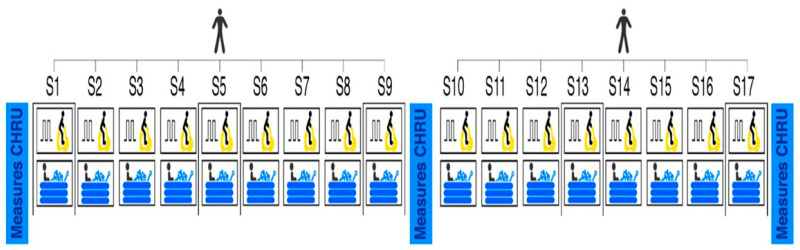
Intervention design. S = session.

**Table 1 ijerph-20-06128-t001:** Participant characteristics and the results of tolerance, safety, and effectiveness measures at baseline and the end of phase I (eighteen patients).

Age (years)	55.1 (11.9)	
Female	17	
Body Mass Index (kg·m^−2^)	25.5 (4.7)	
Work Status—working, *n*	12	
Tabaco consumption, *n* (active/ex-smoker/non-smoker)	2/5/11	
Chronicity of disease (years)	12.6 (9.8)	
Questionnaire SQUASH	5513.5 (2591.0)	
Tolerance measures	Baseline	S9
BORG (points)	11.6 (1.7)	12 (1.4)
VAS-temperature (points)		
First minute	2.9 (1.1)	2.6 (1.1)
Fifth minute	1.9 (0.7)	1.7 (0.7)
Tenth minute	1.7 (0.8)	1.4 (0.6)
Fifteenth minute	1.6 (0.8)	1.4 (0.6)
Safety measures		
Echography (0–64 points)	6.4 (3.8)	4.3 (2.9)
Painful Joints (*n*)	4.3 (5.0)	1.7 (2.8)
Swollen Joints (*n*)	0.9 (2.9)	0.7 (1.0)
Pain (0–100 points)	42.2 (24.4)	38.8 (15.2)
Questionnaires		
HAQ	0.7 (0.5)	0.6 (0.6)
SF-36 physical composite	58 (32.9)	58.4 (33.1)
SF-36 mental composite	59.8 (22.9)	64.1 (24.5)
Effectiveness measures		
Pulse Wave Velocity (m·s^−1^)	8.9 (1.2)	7.0 (0.8) **
Systolic Blood Pressure (mmHg)	120.9 (14.2)	118.2 (13.1)
Diastolic Blood Pressure (mmHg)	68.5 (8.3)	68.6 (10.5)
Heart Rate (bpm)	70.2 (8.4)	66.6 (5.5) *

All results were expressed as the mean (SD). * *p* < 0.05, ** *p* < 0.001, a significant difference from baseline.

**Table 2 ijerph-20-06128-t002:** The results of tolerance, safety, and effectiveness measures at baseline and the end of phase II (eleven patients).

Tolerance Measures	Baseline	S9	S17
BORG (points)	10.9 (1.5)	12.1 (1.6)	11.8 (2.1)
VAS-temperature (points)			
First minute	3 (0.8)	2.7 (1.1)	2 (1.0)
Fifth minute	1.9 (0.7)	1.7 (0.6)	1.3 (0.6)
Tenth minute	1.7 (0.8)	1.5 (0.5)	1.1 (0.3)
Fifteenth minute	1.6 (0.8)	1.5 (0.5)	1.1 (0.3)
Safety measures			
Echography (0–64 points)	7.9 (4.2)	4.9 (3.2)	3.6 (2.3)
Painful Joints (*n*)	3.0 (3.5)	2.2 (3.4)	1.3 (3.1)
Swollen Joints (*n*)	0.7 (1)	0.5 (0.9)	0.5 (1.0)
Pain (0–100 points)	43.6 (20)	40.5 (18.1)	39.5 (22.3)
Questionnaires			
HAQ	0.7 (0.5)	0.8 (0.6)	0.6 (0.4)
SF-36 physical composite	61.4 (32)	56.6 (35.4)	64.1 (33.4)
SF-36 mental composite	59.1 (25)	58.7 (23.9)	59.8 (26.2)
Effectiveness measures	Baseline	S9	S17
Pulse Wave Velocity (m·s^−1^)	9.1 (0.9)	7.0 (0.8) *	6.9 (0.7) *
Systolic Blood Pressure (mmHg)	124.9 (12.1)	121.1 (13.2)	122 (15)
Diastolic Blood Pressure (mmHg)	70.1 (7.8)	69.8 (12.3)	72.2 (9.7)
Heart Rate (bpm)	71.5 (8.5)	66.8 (6.5)	68.3 (8.6)

All results were expressed as the mean (SD). * *p* < 0.05, a significant difference from baseline.

## Data Availability

The data presented in this study are available on request from the corresponding author.

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
