# Peer review of "Effects of an Exercise Program and Cold-Water Immersion Recovery in Patients with Rheumatoid Arthritis (RA): Feasibility Study"

_ijerph, 2023, doi:10.3390/ijerph20126128_

Round 1

Reviewer 1 Report

The paper by Peres et al. addresses a very interesting subject. The authors report the results of a study regarding the role of exercise program and cold-water immersion Recovery in patients with RA. I would like to express sincere gratitude to get the opportunity to review your manuscript. The effort of the author is appreciated. The manuscript is well written, and the study protocols and data scientifically sound and are presented clearly. Congratulations on your results. There is a need to have large multicenter studies and hope to see data in the future. I would recommend addressing the following issues during the revision of the manuscript:

-Line 16 – “patients present is an increased” please assess the writing, “is” I think it should not be.

-Lines 72 and 83 “XXXXX” should be changed to the mane of the hospital where the study was conducted. 

-Please also include the limitations of the study.

Author Response

The paper by Peres et al. addresses a very interesting subject. The authors report the results of a study regarding the role of exercise program and cold-water immersion Recovery in patients with RA. I would like to express sincere gratitude to get the opportunity to review your manuscript. The effort of the author is appreciated. The manuscript is well written, and the study protocols and data scientifically sound and are presented clearly. Congratulations on your results. There is a need to have large multicenter studies and hope to see data in the future. I would recommend addressing the following issues during the revision of the manuscript:

Authors: Thank you for your comments and please find below the answers to the items that you requested the modifications. The alterations in the manuscript are highlighted in red the text.

-Line 16 – “patients present is an increased” please assess the writing, “is” I think it should not be.

Authors: We agree with you. The “is” was deleted.

-Lines 72 and 83 “XXXXX” should be changed to the mane of the hospital where the study was conducted. 

Authors: Thank you. We added the hospital name where the study was conducted.

-Please also include the limitations of the study.

Authors: In accordance with other comments, we added the Limits in a specific paragraph at the end of the discussion (lines 282-290).

Reviewer 2 Report

The main topic of the manuscript is the effect of an exercise and recovery program after immersion in cold water in patients with rheumatoid arthritis.

To the authors.

Did the authors register the study as a clinical trial?

The number of participants is small. The explanation of the sample size calculation is not clear, please explain in more detail and add relevant references.

An important drawback is the lack of a control group.

The authors wrote about the differences between the groups of Phase 1 and Phase 2 participants, please compare them in the table.

Author Response

Answers to comments of Reviewer 2

The main topic of the manuscript is the effect of an exercise and recovery program after immersion in cold water in patients with rheumatoid arthritis.

Authors: Thank you for your comments and please find below the answers to the items that you requested the modifications. The alterations in the manuscript are highlighted in red the text.

To the authors.

Did the authors register the study as a clinical trial?

Authors: Yes, the study was registered under de number NCT03911830 on ClinicalTrials.gov.

The number of participants is small. The explanation of the sample size calculation is not clear, please explain in more detail and add relevant references.

Authors: In accordance with other comments, we explained in more details our sample size calculation and we added the relevant references (lines 166-170).

An important drawback is the lack of a control group.

Authors: We agree with you. A promising design for this study would be to have four groups: the experimental, physical activity, cold-water immersion, and control groups. This would allow the identification of the benefits of proposed therapies, isolated and combined. However, it was necessary to show the feasibility of this project for the rheumatoid arthritis population. Thus, this is a feasibility project that encourages the development of a more robust future study.

The authors wrote about the differences between the groups of Phase 1 and Phase 2 participants, please compare them in the table.

Authors: Sorry if we don’t make it clear in the text. In Table 1, we showed the results of all participants whereas in Table 2, we showed only the participants who had an improved at least 1,5 m.s-2 in PWV values and followed to phase 2 (in accord according to our sample calculation). We compared the results of these participants between their improvement at the end of Phase 1 and at the end of Phase 2. This allowed us to explain the patient’s profile who responds to the therapies. However, in the discussion section, we added a comparison between the improvement of patients during Phase 2 compared to the improvement of patients who participated only in Phase 1 (lines 289-292).

Reviewer 3 Report

Dear Authors,

Manuscript Number: ijerph-2414186

Title Manuscript: Effects of an Exercise Program and Cold-Water Immersion Recovery in Patients with Rheumatoid Arthritis (RA): Feasibility Study

This feasibility study is an interesting topic since the participants and interventions of this study are patients with rheumatoid arthritis and both exercise & cold-water immersion recovery interventions, but at the moment MAJOR REVISIONS are necessary in order to make it suitable for a final decision for “IJERPH”.

This feasibility study is an interesting topic since the participants and interventions of this study are patients with rheumatoid arthritis and both exercise & cold-water immersion recovery interventions, but at the moment MAJOR REVISIONS are necessary in order to make it suitable for a final decision for “IJERPH”.

This study examined an interval exercise intervention (17 sessions) followed by cold water immersion recovery in patients with rheumatoid arthritis. Parameters were assessed before, after 9 and 17 sessions and included evaluation of acceptability by perceived exertion (Borg) and water temperature (VAS) measures at each session; safety by a number of painful and swollen joints (echography), physical function (health assessment questionnaire), general health status (short form–36) measures; and effectiveness by arterial stiffness (pulse wave velocity-PWV) measures. The results showed a good acceptability of the program, no patient dropped out of the protocol or even presented difficulties or perceived pain. The PWV values decreased significantly after nine exercise sessions. No aggravation of symptoms has been noted. 

POINTs of STRENGTH:

1) Effects of both exercise and cold-water immersion recovery interventions in patients with rheumatoid arthritis in a feasibility study;  

POINTs of WEAKNESS (and/or should be revised to improve the manuscript):

Abstract:

2) The purpose of this study is unclear. Please clarify;

3) Please add gender, mean age, weight, BMI of participants as well as type of exercise in the “methods” section of the abstract;

4) The significance level of results is unclear. Please clarify in the results section;

1. Introduction:

5) The hypothesis and purpose of this study can be stated in more detail;

2. Materials and Methods

2.1. Participants

6) The recruitment/screening process of participants OR inclusion and exclusion criteria should be described in more detail such as BMI, physical fitness level/VO2max OR METs, drug interventions OR free of medication, and so on;

2.2. Intervention

7) The time (morning, afternoon or other), place and type of exercise for this study are unknown; please clarify;

8) Considering the important effects of nutrition on rheumatoid arthritis, did the authors monitor the nutritional status of participants during follow-up? IF YES, please add the method of controlling nutritional status;

9) Is there a control group in this study? IF YES, please provide;

2.5. Statistical Analysis

10) The significance level of statistical analysis was considered for one-tailed OR two-tailed? Please clarify;

4. Discussion & 5. Conclusions

11) As mentioned above, authors will agree that the limitations section has to be expanded.

12) Please provide clinical perspectives for this study;

13) What does this feasibility study add to the literature? Please explain;

14) What are the implications for clinical studies?;

References

15) References section is not always in accordance with the authors' guidelines. In particular, please check No. 7, 8, 9, 12, 14 and 28 for validation.

Best Regards

18 May 2023

Author Response

Answers to comments of Reviewer 3

Dear Authors,

Manuscript Number: ijerph-2414186

Title Manuscript: Effects of an Exercise Program and Cold-Water Immersion Recovery in Patients with Rheumatoid Arthritis (RA): Feasibility Study

 This feasibility study is an interesting topic since the participants and interventions of this study are patients with rheumatoid arthritis and both exercise & cold-water immersion recovery interventions, but at the moment MAJOR REVISIONS are necessary in order to make it suitable for a final decision for “IJERPH”.

 Authors: Thank you for your comments and please find below the answers to the items that you requested the modifications. The alterations in the manuscript are highlighted in red the text.

 POINTs of STRENGTH:

1) Effects of both exercise and cold-water immersion recovery interventions in patients with rheumatoid arthritis in a feasibility study; 

POINTs of WEAKNESS (and/or should be revised to improve the manuscript):

Abstract:

2) The purpose of this study is unclear. Please clarify;

Authors: As requested, we have modified the sentence to make our proposal clearer (lines 19).

3) Please add gender, mean age, weight, BMI of participants as well as type of exercise in the “methods” section of the abstract;

Authors: As requested, the gender, mean age, weight, BMI of participants were added (lines 21-22). The type of exercise was also added (lines 20).

4) The significance level of the results is unclear. Please clarify in the results section;

Authors: In accordance with other comments, we added the p-value.

  1. Introduction:

5) The hypothesis and purpose of this study can be stated in more detail;

Authors: In accordance with other comments, we explained in more detail the hypothesis and purpose of this study (lines 70-74).

  1. Materials and Methods

2.1. Participants

6) The recruitment/screening process of participants OR inclusion and exclusion criteria should be described in more detail such as BMI, physical fitness level/VO2max OR METs, drug interventions OR free of medication, and so on;

Authors: Details have been added in the Materials and Methods, Participants section (lines 72-73 and 82-83). It should be noted that weight and fitness level were not part of inclusion-exclusion factors, but were controlled. These parameters could inform us about the patient profiles more susceptible to change at the end of this protocol.

2.2. Intervention

7) The time (morning, afternoon, or other), place, and type of exercise for this study are unknown; please clarify;

Authors: As requested, the time (morning or afternoon) was decided by the participant according to his availability, and maintained for all sessions (see lines 92-93). The place where sessions were realized was added in lines 94-95. The type of exercise is described in the Interval Exercise Program section.

8) Considering the important effects of nutrition on rheumatoid arthritis, did the authors monitor the nutritional status of participants during follow-up? IF YES, please add the method of controlling nutritional status;

Authors: No, we did not monitor the nutritional status. However, all patients had regular medical monitoring during their participation in this study.

9) Is there a control group in this study? IF YES, please provide;

Authors: No, there is not a control group. A promising design for this study would be to have four groups: the experimental, physical activity, cold-water immersion, and control groups. However, it was necessary to show the feasibility of this project for the rheumatoid arthritis population. The actual study allowed us the identification of risks and benefits of physical activity following cold immersion. Thus, this is a feasibility project that encourages the development of a more robust future study.

2.5. Statistical Analysis

10) The significance level of statistical analysis was considered for one-tailed OR two-tailed? Please clarify;

Authors: As it is a feasibility study, we conducted a two-tailed statistical analysis because we were interested to determine if there was a difference, either in favor or against the therapy.

  1. Discussion & 5. Conclusions

11) As mentioned above, the authors will agree that the limitations section has to be expanded.

Authors: We agree with you. In accordance with other comments, we added the Limits in a specific paragraph at the end of the discussion (282-290).

12) Please provide clinical perspectives for this study

Authors: Although the program consisted of moderate-high intensity physical activity, there was no exacerbation of the disease when associated with cryotherapies. Physical exercise controlled by a health professional with the application of cryotherapy proved to be well tolerated and accepted by the patients. However, our study was performed with a small and homogeneous population, not presenting high blood pressure values. Therefore, the application of this program must be done under supervision.

13) What does this feasibility study add to the literature? Please explain;

Authors: We did a systematic review (Peres, 2017)[i] which showed the lack of a protocol in the literature associating physical activity with the application of cryotherapy for RA patients. In addition, cryotherapy protocols have proven to be complex and costly. We showed that it is possible to apply cryotherapy in a simple way, allowing pain and inflammation control after a physical activity of moderate intensity.

14) What are the implications for clinical studies?

Authors: This study encourages us to perform this same protocol more robustly. The current protocol does not allow us to distinguish the benefits of each therapy applied separately. It would also be interesting to choose the subjects in order to have different groups (heterogeneous) in relation to blood pressure (normal and different grades of hypertension) and PWV.

References

15) References section is not always in accordance with the authors' guidelines. In particular, please check No. 7, 8, 9, 12, 14 and 28 for validation.

Authors: As requested, the references were checked and corrected when necessary.

[i] Peres D, Sagawa Y Jr, Dugué B, Domenech SC, Tordi N, Prati C. The practice of physical activity and cryotherapy in rheumatoid arthritis: systematic review. Eur J Phys Rehabil Med. 2017 Oct;53(5):775-787. doi: 10.23736/S1973-9087.16.04534-2. Epub 2016 Dec 19. PMID: 27996221.

Round 2

Reviewer 2 Report

The corrections made have sufficiently improved the manuscript. This will allow the publication of this article.

Reviewer 3 Report

Dear Authors,

Manuscript Number: ijerph-2414186

Title Manuscript: Effects of an Exercise Program and Cold-Water Immersion Recovery in Patients with Rheumatoid Arthritis (RA): Feasibility Study

I am very grateful to the authors for their efforts.

In general, this manuscript has found suitable content after correcting major revisions, and the modified revisions are accepted.

However, this manuscript suffers from poor grammar- please organize a proper proof read by a native-speaking person before the final version. For example, (presents OR present!?) "the RA patients presents usually associated pathologies, such as hypertension.", and so on. 

Best Regards

6 June 2023

 This manuscript suffers from poor grammar- please organize a proper proof read by a native-speaking person before the final version. For example, (presents OR present!?) "the RA patients presents usually associated pathologies, such as hypertension.", and so on.